# Whole-Genome Resequencing Reveals Selection Signal Related to Sheep Wool Fineness

**DOI:** 10.3390/ani13182944

**Published:** 2023-09-16

**Authors:** Wentao Zhang, Meilin Jin, Taotao Li, Zengkui Lu, Huihua Wang, Zehu Yuan, Caihong Wei

**Affiliations:** 1State Key Laboratory of Animal Biotech Breeding, Institute of Animal Sciences, Chinese Academy of Agricultural Sciences, Beijing 100193, China; m18251871965@163.com (W.Z.); jmlingg@163.com (M.J.); ltt_ltt2020@163.com (T.L.); wanghuihua@caas.cn (H.W.); 2Key Laboratory of Animal Genetics and Breeding on Tibetan Plateau, Ministry of Agriculture and Rural Affairs, Lanzhou Institute of Husbandry and Pharmaceutical Sciences, Chinese Academy of Agricultural Sciences, Lanzhou 730050, China; luzengkui@caas.cn; 3College of Animal Science and Technology, Yangzhou University, Yangzhou 225000, China; yuanzehu@yzu.edu.cn

**Keywords:** wool fineness, sheep, selection signal, whole-genome resequencing, Fst and θπ Ratio and XP-EHH

## Abstract

**Simple Summary:**

Wool is a very important agricultural product and an ideal raw material for the production of high-quality textiles and clothing. In addition, the wool industry provides important support for economic development and the livelihoods of rural communities, creating employment opportunities and sources of income for farmers. Whole-genome sequencing technology has important applications in the study of genetic diversity, population structure, and selection pressure in the sheep genome. In this study, we analyzed the population structure and genomic differences of eight breeds. In addition, we identified a series of candidate genes that may be related to hair follicle development, wool traits, lipid metabolism, and androgen metabolism. This study provides valuable genomic resources and a theoretical basis for future wool improvement.

**Abstract:**

Wool fineness affects the quality of wool, and some studies have identified about forty candidate genes that affect sheep wool fineness, but these genes often reveal only a certain proportion of the variation in wool thickness. We further explore additional genes associated with the fineness of sheep wool. Whole-genome resequencing of eight sheep breeds was performed to reveal selection signals associated with wool fineness, including four coarse wool and four fine/semi-fine wool sheep breeds. Multiple methods to reveal selection signals (Fst and θπ Ratio and XP-EHH) were applied for sheep wool fineness traits. In total, 269 and 319 genes were annotated in the fine wool (F vs. C) group and the coarse wool (C vs. F) group, such as *LGR4*, *PIK3CA*, and *SEMA3C* and *NFIB*, *OPHN1*, and *THADA*. In F vs. C, 269 genes were enriched in 15 significant GO Terms (*p* < 0.05) and 38 significant KEGG Pathways (*p* < 0.05), such as protein localization to plasma membrane (GO: 0072659) and Inositol phosphate metabolism (oas 00562). In C vs. F, 319 genes were enriched in 21 GO Terms (*p* < 0.05) and 16 KEGG Pathways (*p* < 0.05), such as negative regulation of focal adhesion assembly (GO: 0051895) and Axon guidance (oas 04360). Our study has uncovered genomic information pertaining to significant traits in sheep and has identified valuable candidate genes. This will pave the way for subsequent investigations into related traits.

## 1. Introduction

The wool fineness refers to the diameter of the sheep’s wool fibers, and it is an important indicator of the quality of the wool. Different sheep breeds vary greatly in their wool fineness; the smaller the fineness, the softer the wool fiber. Coarse wool can therefore be used to make heavy clothing and blankets, while fine wool can be used to make soft, gentle clothing, such as knitwear and textiles [1]. In the 1990s, Australia had bred sheep towards the fine and ultra-fine types, and wool under 20 microns had reached 30% of the total production in 2000. Although China has a large indigenous sheep breed, the lack of efficient breeding has resulted in relatively low yields and inconsistent quality of fine wool. China has a large market demand for fine wool but has long relied on imports. The selection signal analysis has revealed candidate genes related to wool fineness and provided support for improving wool fineness in sheep.

Genome-wide association analysis and whole-genome resequencing techniques have allowed the identification of genes and SNPs associated with wool fineness. In Chinese Merino sheep, the roles of *TSPEAR*, *PIK3R4*, and *KRTCAP3* in hair follicle development and fiber diameter have been suggested [2], and the *EPHA5* gene was significantly associated with wool curl traits [2]. A study comparing the skin of coarse and fine wool sheep revealed the potential effectors (*APCDD1*, *FGF20*, *DKK1*, *IGFBP3*, and *SFRP4*) that regulate the skin compartments of primary wool follicles to shape the variable wool fiber thickness [3]. *BNC1* was found to be significantly associated with mean fiber diameter (MFD) in sheep [4]. Among the type II irs-specific keratins, *KRT71* was expressed in all three irs layers, but only *KRT71* was expressed in the outer Henle layer [5,6]; *KRT72* and *KRT73* were exclusively expressed in the inner cuticle, while *KRT74* expression was restricted to the middle Huxley layer [6,7]. Another study identified some genes belonging to KAPs and KIFs that may affect the fineness of wool [8]. Overexpression of mutant C-KRT74 likely enlarges Huxley cells by enhancing the KIF network, which in turn allows the hair shaft to be shaped more finely [9]. Mutations upstream of *EDAR* increase placode formation [9]. A non-synonymous variant of *EDAR* is associated with hair thickness in Asian populations [10]. Among the *BAAT* genes of Liaoning cashmere goats, the TT genotype at locus T7967C had the best cashmere fineness [11]. *GHR* is progressively downregulated during hair follicle development and correlates with curl number (CN) [12]. Furthermore, it also demonstrates a strong negative correlation between fiber diameter and follicle density [13]. These genes are implicated in processes that regulate wool cell development and fiber morphology and play an important role in regulating wool fineness. By exploring the molecular mechanisms and key genes of wool cell differentiation through single-cell RNA sequencing, it was found that the WNT signaling pathway [14,15,16], TGFβ signaling pathway [16,17], fibroblast growth factor (FGF) signaling pathway [18], and hedgehog signaling pathway [17] play important regulatory roles during cell differentiation and fiber formation. The Wnt/beta-catenin and EDA/EDAR/NF-kappaB signaling pathways play a role in the initiation and maintenance of the primary hair follicle placodes [19,20]. The hedgehog signaling pathway regulates the function of the mesenchymal niche in the hair follicle by means of the SCUBE3/TGF-β mechanism [17]. The EBF1/WNT10A interaction is highly significant in relation to hair formation during the anagen phase [21]. The transcription factor FOXO1 is associated with the growth and differentiation of dermal papillae (DP) and hair follicle epithelial cells [18,22]. The WNT/β-catenin pathway’s differential activity in dermal papilla cells (DPCs) may have an effect on the curl of the wool [23]. EDA signaling has been shown to play an essential role in regulating hair curl [10,24]. Such key factors and signaling pathways can regulate and influence the morphology and quality of wool fibers, and, consequently, the fineness. It was discovered that subcutaneous fat cells play an important role in the quality control of wool. The nuclear transcription factor FOXO1 reduces androgen-induced androgen receptor (*AR*) target gene expression [25] to inhibit the AR [26]. Lack of nuclear FOXO1 increases the transactivation of the AR [22,25] and alters the activity of critical nuclear receptors and key genes, which are implicated in the proliferation of pilosebaceous keratinocytes, sebaceous lipogenesis, and the expression of perifollicular inflammatory cytokines [22]. During the regulation of wool development and fiber morphology, subcutaneous adipocytes secrete several factors that can affect wool fineness. From the above research progress on the genes, pathways, and mechanisms related to wool fineness, it can be found that there are many studies on fine wool genes, but they can only explain a certain proportion of the variation in wool fiber thickness, and there is still a lack of research on the regulatory mechanisms. Therefore, there is still a need to discover more genes related to fine wool formation. The study of coarse wool is relatively rare. Coarse wool is also one of the important components of wool resources, but it lacks commercial value today, and its conservation needs to be valued. The difference between modern wool carpets and ancient oriental wool carpets shows a lack of a wide variety of high-quality coarse-fiber-diameter wool. Therefore, there is also some value in studying the coarse-wool-related genes that diversify coarse wool varieties [27].

While other economic traits in sheep have been more intensively studied, the potential genes regulating wool fineness have not yet been explored, and the underlying genetic mechanisms of wool fineness remain unclear. Upon summarizing the above, we still deem it necessary to search for new candidate regulatory genes. We therefore performed a whole-genome resequencing to reveal the selection signatures of sheep wool fineness, including four coarse wool and four fine/semi-fine wool sheep breeds. First, the genetic diversity between populations was calculated using the Pi [28] method, and then the degree of genetic differentiation between populations was calculated using the Fst [29,30] method. Finally, the XP-EHH [31] method was used to identify genome-wide natural selection events and compare them with differentiation and diversity results to identify selection signals associated with sheep wool fineness. These resulted in the identification of candidate genes associated with sheep wool fineness, as well as supporting research into potential genetic mechanisms of sheep wool fineness.

## 2. Materials and Methods

### 2.1. Ethics Statement

All experimental procedures involving sheep were approved by the Animal Ethics Committee of the Institute of Animal Sciences, Chinese Academy of Agricultural Sciences (IAS-CAAS).

### 2.2. Sample Collection and Sequencing

Blood samples were collected from eight sheep breeds (Table 1), including Australian Merino sheep (AME), German Mutton Merino sheep (GME), Tong sheep (TON), Hanzhong sheep (HZS), Lop sheep (LOP), Hu sheep (HUS), Small Tailed Han sheep (STH), and Ujimqin sheep (UJI), using jugular vein blood collection. Sequencing libraries were generated using the Truseq Nano DNA HT Sample Preparation Kit (Illumina, San Diego, CA, USA). The whole genomes of the 28 sheep were sequenced on the Illumina Hiseq 2000 platform. The resequenced data were used for subsequent analysis.

We selected healthy, breed-typical, 1-year-old individuals to ensure a representative and comparable group of animals (whenever possible, full/half siblings were used to minimize individual differences). The farm provides high-quality feed and adequate clean water; a suitable and comfortable feeding environment, including appropriate space, dry beds, and a suitable temperature range; regular health checks and vaccination programs; hygiene maintenance, reasonable exercise and rest, and breeding management; and regular inspection and maintenance of equipment.

### 2.3. Alignments and Quality Control

The raw reads of fastq format were first processed through a series of quality control procedures using FastQC to ensure reliable reads in the subsequent analyses. The standards of quality control were followed, including: (1) removing reads with ≥10% unidentified nucleotides (N); (2) removing reads with >20% bases having phred quality less than 5; (3) removing reads with >10 nt aligned to the adapter, allowing ≤10% mismatches; and (4) removing putative PCR duplicates generated by PCR amplification in the library construction process (reads 1 and 2 of 2 paired-end reads that were completely identical).

Valid high-quality sequencing data were compared to the reference genome (GCF_016772045.1_ARS-UI_Ramb_v2.0) by BWA (v 0.7.17) [32] software (parameter: mem -t 4 -k 32 -M); the results were matched by SAMTOOLS [33] to remove duplicates (parameter: rmdup). Single Nucleotide Polymorphism (SNP) is a type of genetic variation occurring at a single nucleotide position in the DNA sequence. To improve the accuracy of data processing, SNPs were screened to pass the following criteria: (1) the number of SNPs supported (depth of coverage) was above 2; (2) the proportion of MIS (deletions) was less than 10%; and (3) the MAF (minimum allele frequency) was more outstanding than 5% [34].

### 2.4. Population Structure Analysis

Prior to analysis, all SNPs were trimmed with PLINK 1.09 [35] software’s indep-pairwise [36], with parameters set to a non-overlapping window of 25 SNPs, a step size of 5 SNPs, and a threshold of 0.05 for r^2^ to obtain independent SNP markers. To understand the clustering of the population, principal component analysis (PCA) was carried out using PLINK 1.09 [35]. In order to understand the distance between relatives, we constructed neighbor-joining (N-J) trees [37] using MEGA (v 7.0) software [38] and visualized them using ITOL (v 6) software [39] (https://itol.embl.de/upload.cgi (accessed on 7 June 2023)). To observe the degree of segregation of the population, as well as to validate the results of PCA and N-J trees, we constructed the population genetic structure using ADMIXTURE (v 1.3) software [40], with k values set from 2 to 9. Then, we used the CV error to detect the optimal K.

### 2.5. Selection Signal Analyses

Before the analysis, AME, GME, TON, and HZS were clustered as F and LOP, HUS, STH, and UJI as C. This study first used the population differentiation index (Fst [30]) and the nucleotide diversity ratio (θπ Ratio) method, which is a very potent method for detecting regions of selective elimination. Fst has been widely used to identify selective signals [30] in genome-wide SNPs [41]. θπ indicates the nucleotide polymorphism; the higher the degree of selection, the lower the polymorphism. The stronger selection signal was screened jointly by comparing Fst and θπ of F and C to facilitate the screening of target genes. The Fst and θπ values were calculated using VCFtools (0.1.15) software [42] with the parameters: -fst-window-size 50,000 -fst-window-step 50,000, followed by θπ Ratio [28]. The loci within the window where both the Fst and θπ Ratio were the top 5% were extracted as significant SNP loci (namely, the candidate loci for the selection signal).

The XP-EHH [31] method is to detect natural selection. Extended Haploid Homozygosity (EHH) is a metric used in population genetics to assess the degree of selection pressure on haplotype homozygosity and genomic regions. The method is based on Single Nucleotide Polymorphism (SNP) Extended Haploid Homozygosity (EHH). It identifies selection signals for population genetic variation by comparing frequencies and EHH values across populations. It will better use single-population information and more efficiently discover candidate regions and genes within populations. We used fastphase 1.4 [43] to estimate haplotypes, using population marker information to estimate the haplotype of each chromosome with the options set to: -Ku40 -Kl10 -Ki10. In order to determine whether there had been selection in the experimental population, the XP-EHH values were calculated using the haplotype information in the XP-EHH programme from http://hgdp.uchicago.edu/Software/ (accessed on 7 June 2023). Subsequently, XP-EHH values were calculated using the sliding window method, with the window size set to 50 kb and the step size set to 20 kb. Then, the average of all SNPs in each sliding window was calculated. The loci within the window with XP-EHH of the top 5% were extracted as significant SNP loci, meaning they were candidates for the selection signal. XP-EHH detects whether the locus is pure in one population and has polymorphism [31] in the other by comparing the EHH score of a core SNP between two populations. A negative XP-EHH score indicates that selection occurred in the reference population, while a positive XP-EHH score indicates that selection occurred in the experimental population.

The Fst [30] method calculates the Fst from the variation in allele frequencies between populations, and the higher the Fst value tends to be towards 1, the greater the level of population differentiation [44]. However, as false positives can occur with the method based on unit point SNPs to find potentially selected regions, the sliding window approach was chosen to reduce the probability of false positives in order to improve accuracy [45]. The θπ Ratio [28] is the ratio between two populations calculated from gene heterozygosity. The more the θπ Ratio deviates from 1, the higher the level of genomic selection [46]. The XP-EHH method [31] was introduced because of the large variation in the number of individuals in the sample, while the XP-EHH method is less affected by the number of individuals in the sample. XP-EHH is a method based on the homozygosity of haplotypes on the genome to identify the regions where selection occurs between different populations, which provides a stronger data support for in-depth screening of candidate genes [47]. Three methods (FST, θπ Ratio, XP-EHH) are based on comparing the genetic divergence index, nucleotide diversity (π-value), and haploid knockout elongation in two populations, respectively, to look for signals of selection associated with specific traits. Higher FST values indicate greater genetic divergence between populations and can be inferred to signal the presence of selection. Higher values of θπ Ratio indicate that the loci in the study population may have been affected by positive selection during evolution. A higher value of XP-EHH implies that there may be a signal of selection in favor of the trait in the study population.

Three methods were used in this study to validate each other: the θπ Ratio method to provide information on gene diversity, the Fst method to provide information on the degree of gene differentiation, and the XP-EHH method to provide information on natural selection events, which together can provide more comprehensive and accurate information on genetic evolution [48,49].

### 2.6. Detection and Annotation of Candidate Genes

The Fst and θπ Ratio and XP-EHH screened the top 5% of the candidate SNP loci as the “outlier loci” for this experiment. In total, 50 kb upstream and downstream of the SNP loci were considered selection signaling regions. ANNOVAR [50] was used to annotate genes with the sheep reference genome. The Venn diagram is based on the candidate genes obtained from the Fst and θπ Ratio and XP-EHH.

### 2.7. Candidate Gene Enrichment Analysis

Functional enrichment of candidate genes was performed using DAVID 6.8 [51] (https://david.ncifcrf.gov/ (accessed on 9 June 2023)), with Gene Symbol as the input parameter and Ovis_aries selected for the background organism. We counted the number of genes associated with these GO terms. We calculated the significance of their enrichment using the hypergeometric distribution test with *p*-value < 0.05 as a significant enrichment result. These genes were also enriched for KEGG analysis using Kobas 3.0 [52] (http://kobas.cbi.pku.edu.cn/kobas3/genelist/ (accessed on 9 June 2023)), with *Ovis_aries* selected for the background organism, and the hypergeometric test/Fisher’s exact test was used as the statistical method, selecting *p*-value < 0.05 as a significant enrichment result.

## 3. Results

### 3.1. Genetic Variation and Population Genetic Analysis

In this study, we performed whole-genome resequencing on 28 individual sheep, generating an average of 7.6× coverage data. After aligning to the reference genome (ARS-UI_Ramb_v2.0), 9,581,315,830 reads were obtained with a coverage rate of 98.03%. Subsequently, variant calling and quality control identified a total of 22,133,207 SNPs. The Single Nucleotide Polymorphisms (SNPs) statistical analysis revealed that the variants predominantly occurred in the intergenic region, followed by the intronic and exonic regions. Among these, the intronic variants comprised 82,379 non-synonymous SNPs and 68,110 synonymous SNPs (Table 2). This comprehensive SNP dataset is valuable for advancing sheep biological research and breeding.

In order to understand the genetic relationships and differences between sheep populations of different hairiness from a genome-wide perspective, PCA, phylogenetic tree construction, and population structure analysis were performed on the eight sheep populations using the obtained SNP datasets.

As shown by the PCA results (Figure 1b), PC1 was able to explain 6.86% of the genetic variation that distinguished the fine wool sheep (AME and GME) from all groups; in the negative direction of PC1, the semi-fine wool and coarse wool sheep (HZS, TON, LOP, HUS, STH, and UJI) clustered together. Combining PC1 with PC2 (5.61% genetic variation) reveals a slight separation between the two semi-fine wool sheep breeds. As indicated by the N-J tree (Figure 1d), the fine wool sheep group (AME, GME) was on a primary branch, whereas the semi-fine and coarse wool sheep (HZS, TON, LOP, HUS, STH, and UJI) were on a major branch. The population genetic structure was constructed using ADMIXTURE (v 1.3) software to confirm the accuracy of the results obtained from PCA and N-J trees (Figure 1c). With K = 2, the fine wool sheep breed was predominantly on a blue background, with a clear transition from the fine wool to the semi-fine and coarse wool sheep breeds. This result is consistent with the PCA and the N-J tree. When K = 3, the HZS segregates from the semi-fine wool sheep population.

### 3.2. Selection Signaling Analysis

We calculated the Fst and θπ Ratio within a 50 kb window (50 kb step size) and screened the shared window for the top 5% of Fst and the top 5% of θπ Ratio as the final candidate window. In total, 2628 candidate SNP loci associated with fine wool formation (F vs. C) and 4020 candidate SNP loci associated with coarse wool formation (C vs. F) were screened (Figure 2a, Appendix A). We annotated 428 candidate genes associated with fine wool formation (F vs. C) and 603 candidate genes associated with coarse wool formation (C vs. F).

Using the top 5% of XP-EHH values as candidate regions for gene annotation (Figure 2b,c), 7035 candidate SNP loci associated with fine wool formation (F vs. C) and 5651 candidate SNP loci associated with coarse wool formation (C vs. F) were screened (Appendix A). After annotation by ANNOVAR software [50], we eventually obtained 1039 and 749 candidate genes associated with fine wool formation (F vs. C) and coarse wool formation (C vs. F) within the top 5% of XP-EHH.

A Venn diagram (Figure 2d,e) based on candidate genes obtained in Fst and θπ Ratio and XP-EHH yielded 269 overlapping genes associated with fine wool formation (F vs. C) and 319 overlapping genes associated with coarse wool formation (C vs. F) (Appendix A).

Based on gene function and previous studies, genes related to hair follicle development, androgen, and fine wool traits, such as *LGR4*, *PIK3CA*, *SEMA3C*, *SOX5*, *DDB2*, *NR1H3*, *SOX6*, *GRIA2*, *RARB*, *JMJD1C*, and *DSG2*, were finally screened in the F vs. C group; *NFIB*, *OPHN1*, *THADA*, *EZH2*, *HSD17B2*, *AR*, *SEMA3D*, *IL1R2*, *DROSHA*, *MSRB3*, *GLI3*, and *KATNAL1* related to hair follicle development, androgen, and coarse wool traits were screened in the C vs. F group.

### 3.3. Enrichment Analysis

Candidate genes screened in the sheep genome using Fst and θπ Ratio and XP-EHH went for GO and KEGG enrichment. First, in the fine wool trait (F vs. C), there are 15 GO significant terms (Appendix A), including 8 significant biological processes, 4 significant cellular components, and 3 significant molecular functions (*p* < 0.05, Figure 3a), including protein localization to plasma membrane (GO:0072659, *IKBKB*, *TSPAN15*, *FAM120C*, *ITGB1BP1*, and *ANK1*), homophilic cell adhesion via plasma membrane adhesion molecules (GO:0007156, *KIRREL3*, *DSG2*, *DSG3*, *DSG4*, and *CDH18*), positive regulation of G1/S transition of mitotic cell cycle (GO:1900087, *ADAM17*, *RRM2*, and *CPSF3*), endoplasmic reticulum and ubiquitin-protein transferase activity (GO:0005783, *GRIA1*, *FADS3*, *CPED1*, *HACE1*, *PEX3*, *ITGA8*, *TMTC2*, *ATP10B*, *ACO1*, *ELAVL1*, and *TP63*), and regulation of GTPase activity (GO:004308, *BCL6*, *RAB3GAP2*, *ITGB1BP1*, and *SBF2*). There were also 21 significant GO terms (Appendix A) identified in the coarse wool trait (C vs. F), including 9 significant biological processes, 4 significant cellular components, and 8 significant molecular functions (*p* < 0.05, Figure 3c), including focal adhesion assembly (GO:0048041, *PEAK1*, and *ARHGAP6*), protein localization to plasma membrane (GO:0072659, *IFT20*, *TSPAN15*, *ROCK2*, *TTC7B*, and *TTC7A*), protein localization to plasma membrane (GO:0072659, *IFT20*, *TSPAN15*, *ROCK2*, *TTC7B*, and *TTC7A*), regulation of platelet-derived growth factor receptor–alpha signaling pathway (GO:2000583, *IFT20*, and *CBLB*), CMP catabolic process (GO:0006248, *DPYD*, and *UPB1*), chloride transport (GO:0006821, *GABRA1*, *GLRA3*, *GABRA3*, and *GABRE*), and GTPase activator activity (GO:0005096, *OPHN1*, *RASA2*, *GDI2*, *ARHGAP29*, *SRGAP2*, and *ARHGAP6*).

Then, in the fine wool trait (F vs. C), 38 significant KEGG enrichment pathways were identified (*p* < 0.05, Figure 3b, Appendix A), including Regulation of actin cytoskeleton (oas04810, *ITGA8*, *PIP5K1B*, *PIK3CA*, *PPP1R12A*, and *DIAPH3*), Phosphatidylinositol signaling system (oas04070, *PLCB2*, *INPP1*, *PIP5K1B*, *SYNJ1*, *PIK3CA*, and *MTMR7*), Regulation of lipolysis in adipocytes (oas04923, *PIK3CA*, *PTGS1*, and *GNAI3*), Sphingolipid signaling pathway (oas04071, *PIK3CA*, *PLCB2*, *PPP2R3A*, *GNAI3*, and *SGMS1*), Cortisol synthesis and secretion (oas04927, *PLCB2*, *KCNK2*, and *PBX1*), and PI3K-Akt signaling pathway (oas04151, *ITGA8*, *SYK*, *PIK3CA*, *PPP2R3A*, *MTCP1*, *IKBKB*, *MAGI2*, and *EIF4B*). In the coarse wool trait (C vs. F), KEGG pathway enrichment analysis identified 16 significant KEGG enrichment pathways (*p* < 0.05, Figure 3d, Appendix A), including Axon guidance (oas04360, *SEMA3D*, *ROBO1*, *EFNA5*, *NTN4*, *ROCK2*, *SRGAP2*, *BMPR1B*, *SEMA4G*, and *PAK3*), Regulation of actin cytoskeleton (oas04810, *PDGFD*, *DIAPH2*, *DIAPH3*, *PIP5K1B*, *ROCK2*, *FGF18*, *PIP4K2A*, *FGF2*, and *PAK3*), MAPK signaling pathway (oas04010, *RASGRP1*, *PDGFD*, *EFNA5*, *FGF18*, *MAP2K5*, *NLK*, *FGF2*, and *RASA2*), Arginine and proline metabolism (oas00330, *MAOB*, *MAOA*, and *L3HYPDH*), beta-Alanine metabolism (oas00410, *UPB1*, *ALDH6A1*, and *DPYD*), Pantothenate and CoA biosynthesis (oas00770, *UPB1*, and *DPYD*), Phenylalanine metabolism (oas00360, *MAOB*, and *MAOA*) and Gap junction (oas04540, *PRKG1*, *TUBA8*, *PDGFD*, and *MAP2K5*).

## 4. Discussion

### 4.1. Genetic Variation and Population Genetic Analysis

A coverage of 98.03% indicates that the sequencing experiment obtained a large amount of high-quality sequencing data that cover the bases of the reference genome to a large extent. High coverage reduces the possibility of possible misses and errors and provides more comprehensive and accurate genomic information. In addition, sequencing data with high coverage can provide more confidence and interpretability, which is especially important for annotating variants, searching for mutation sites, and analyzing genomic structure and function. Therefore, the data can be used for subsequent population structure analysis and selection signal analysis. The TS/TV ratio evaluates the magnitude of resequencing errors, which is used to evaluate the quality of SNPs and also reflects the ratio of pure and heterozygous SNPs in species. In our study, the TS/TV ratio was 1.9, close to 2, from which it can be inferred that the distribution of SNPs within the population is relatively balanced and reflects that the genomic population structure tends to be normalized. This provides reliable basic data for subsequent analysis of population structure and selection signals. We can further utilize these data to study aspects of population genetic differences and potential natural selection signals.

In this study, there exists a limited number of breed samples, and the selection signal analysis may be affected by individual differences. Thus, this paper selects representative samples by orientation to minimize the differences between individuals; in addition to this, the joint analysis method can mitigate the effects and errors of individual differences brought about by the small number of samples and compensate for the effects of the insufficient number of samples to a certain extent [31,53,54]. FST, θπ Ratio, and XP-EHH methods have different advantages in the study of genetic differences among sheep populations, selection pressure, etc. [44,46,47]. FST, θπ Ratio, and XP-EHH methods all capture selection signals in different contexts, and for the filtered regions with stronger predicted impacts, the validation experiments can be repeated on them to increase the credibility of the selection signals and reduce the number of occurrences of false positives and false negatives [31,49,53,55]. Thus, the combination of the three methods in this study can cover a wider and more comprehensive genomic region and improve the efficiency and reliability of selection signal screening. Wool fineness has a similar genetic basis in different breeds with low to medium heritability [56]; thus, genes associated with wool fineness may be found in regions of low genetic diversity. Therefore, in this study, the threshold of selection signal analysis was adjusted to the top 5% to avoid missing candidate genes related to wool fineness [57].

In this study, we performed whole-genome resequencing on 28 sheep samples. The sequencing data were compared with the sheep reference genome. Various filtering methods (deep filtering, deletion rates, minimum allele frequencies, etc.) were used to call SNPs from individual sheep to ensure that the final data of high-quality SNPs were obtained. According to the PCA and ADMIXTURE results, the Tong sheep is close to the coarse wool breed and is clearly grouped with the coarse wool breed, while the Hanzhong sheep is farther away from the Tong sheep and very different from the other breeds. This may be due to historical genetic reasons. The Tong sheep, as well as several other coarse-wooled sheep breeds, belong to the Chinese Mongolian/Kazakh line of sheep. These two strains of sheep interact with each other extensively. Due to the special geographic location of the Hanzhong region of China, the Hanzhong sheep has maintained a relatively independent breeding environment and genetic background for a long time. This breed may not have been genetically exchanged as frequently as other more widely distributed sheep breeds. As a result, genetic differences between the Yunnan and Mongolian/Kazakh sheep lines remain.

### 4.2. Annotation of Candidate Genes

There are three main areas of research into wool fineness: the association of genes with wool fineness, the study of wool cell differentiation and development, and the study of mechanisms regulating wool quality. Some of the reported genes are listed to corroborate the previous studies and reflect this paper’s accuracy. Among these, we identified some candidate genes that have been reported to be associated with hair follicle development, wool traits, lipid metabolism, and androgen metabolism. After three methods of selective signaling analysis, the F vs. C (fine wool formation) group annotated 269 candidate genes in overlapping regions. *LGR4* plays a vital role in hair follicle development by activating hair follicle stem cells and influencing the activity of multiple signaling pathways known to regulate hair follicle stem cells to promote a normal hair cycle [16]. In *Lgr4*-deficient mice, hair follicle development was partially impaired, and the expression of *Edar, Lef1*, and *Shh*, which are essential for hair follicle morphogenesis, was reduced in the epidermis [58]. *PIK3CA* expression was lower in superfine wool sheep than fine wool sheep, and its low expression correlated with lower wool fiber diameter [59]. The neuromuscular junction-like structure of the lanceolate complex, a particular structure that enables hair to perceive environmental stimuli, is formed by nerve endings and terminal Schwann cells (TSC) that specifically and densely surround the hair follicle in the isthmus region [60]; *SEMA3C* is closely spatially associated with the lanceolate complex so that it may play a role in the establishment/maintenance of the lanceolate complex in the hair follicle [60]. *SOX5* is involved in several biological pathways of hair follicle development in velvet goats and may be associated with wool characteristics in Inner Mongolian velvet goats [61]. Expression of *DDB2* in non-pigmented supra-hair sheaths (HS) and hair bulbs (HB) is associated with the growth of unpigmented hair fibers [62]. *DDB2* was identified as a novel AR interacting protein [63,64] that mediates contact with AR and the CUL4A-DDB1 complex for AR ubiquitination/degradation [64]. *NR1H3* (*LXR-a*) is present in human sebocytes, and LXR agonists inhibit hair follicle growth [65]. The *GRIA2* gene is associated with hind leg wool traits in goats [66]. The *Sox* family is expressed in early hair follicle development, and network predictions suggest that the MSTRG.223165-miR-21-SOX6 axis is involved in hair follicle development [67]; the targeting relationship between miR-21 and *SOX6* and MSTRG.223165 has been validated in a dual luciferase assay [67]. The transcription factor *RARB* stimulates the promoters of genes involved in androgen production (*StAR*, *CYP17A1*, and *HSD3B2*) and boosts the production of androstenedione, thereby regulating the biosynthesis of androgens [68]. *JMJD1C*, which encodes a histone demethylase, is a coactivator of the AR [69], and its locus on 10q21 affects serum androgen levels [70]. The expression of *DSG2* was correlated with the differentiation status of defined populations within the hair follicle [71]. *DSG2* was highly expressed by the lowest differentiated cells of the cutaneous epithelium, including the fetal and adult hair follicle bulge, the matrix cells of the bulb, and the basal layer of the outer root sheath [71]. However, low levels were found in the basal layer of the interfollicular epidermis [72].

In total, 319 candidate genes were annotated in overlapping regions in the C vs. F group (coarse wool formation). We uncovered a key role for the transcription factors (TFs) nuclear factor IB (*NFIB*) in the maintenance of stem cell (SC) identity: without the NFI TFs, SCs lose their ability to regenerate hair [73]. NFI is a potent regulator of AR-mediated gene expression [74] and negatively regulates the promoter of the *AR* gene [75]. *Lnc-OPHN1-5*, which is physically close to the *AR* gene on the X chromosome, affects androgen synthesis by decreasing AR mRNA translation and protein synthesis [76]. The impact of SNPs in or near *THADA* on hair morphology is associated with curly hair in people of European descent [10]. The dermal PRC2 (Polycomb Repressive Complex 2)—EZH2 activity is important for the initiation of primary hair follicles and is required for the regulation of the NOGGIN-mediated gene network for the initiation of secondary hair follicles [14]. *EZH2* increases serine/threonine kinase 40 (STK40) expression through downregulation of miR-22, thereby accelerating the growth and differentiation of hair follicle stem cells (HFSCs) [77]. MiR-214 targets the *EZH2* and Wnt/β-catenin signaling pathway to regulate human HFSC proliferation and differentiation [15]. The hair follicle has autonomous control over androgen metabolism by adjusting steroid hormone production and degradation according to local requirements [78]. The presence of type 2 *17β-HSD* in the root sheath cells of the hair follicle is associated with the maintenance of androgen homeostasis in the hair follicle [78], and several studies have suggested that it is associated with androgen inactivation [79,80] or effective decreases in production [81]. *SEMA3D*, which participates in several biological pathways of hair follicle development in velvet goats, may be related to wool traits in Inner Mongolia velvet goats [61]. Androgens, such as testosterone and DHT, are vital steroid hormones in the body, and AR is the receptor that androgens must rely on to function [78]. Androgens act on hair follicles via the AR to cause follicle miniaturization and inhibit hair growth [82]. In human hair follicles, *AR* expression is restricted to DPC (dermal papilla cells) and was not found in the outer root sheath (ORS), hair bulb, or bulge [83]. *DROSHA* inhibits DNA damage in rapidly proliferating hair follicle stromal cells, promotes hair follicle development, and maintains hair follicle structure and its associated stem cells [84]. Sonic Hedgehog(*Shh*) promotes the proliferation of epidermal progenitors required for embryonic hair follicle development through transcriptional activation of N-myc and c-myc, and this transcriptional regulation is controlled by the balance of *Gli* activator and repressor functions [85]; additional roles for *Gli3* in the negative regulation of cell proliferation occurs via the inhibition of CyclinD2 and N-myc transcription [85]. HH/GLI signaling, which is essential for hair follicle development and morphogenesis [86,87,88], is a signaling network consisting of the HH signaling pathway and the GLI signaling pathway, and the HH signaling pathway can transmit its signals by activating members of the *GLI* family. GLI/EGF target gene *IL1R2* is expressed explicitly in ORS [89]. Whereas *IL1R2* and *EGFR* are co-expressed in ORS, the synergistic effect of HH/GLI and EGFR signaling determines the development of ORS cells [89]. *MSRB3* is involved in wool physiology [90,91]; SelR/MsrB reverses Mical oxidation of actin to restore its normal polymerization properties [90,91], and actin bundles in hair follicles, similar to stress fibers, act as a tensile scaffold for hair follicle growth [92]. The candidate gene *KATNAl1*, which affects fiber fineness, was identified in the OAR10 region by comparing the Afec-Assaf strain with its parent Awassi variety [93].

Through the studies of Wu et al. [57] and Ma et al. [94], it was also confirmed that *MYO3B* and *MACROD2* genes were associated with hair fineness and *FST* genes were associated with wool density in this study. Combined with the studies of Wu et al. [57] and Ma et al. [94], we can learn that although *KRT* and *KAP* genes are the main structural proteins of animal hair, the expression of these genes is relatively low in the finer wool. This may be part of the developmental and structural regulatory mechanism of finer wool. However, the study by Liang et al. [9] reveals a special case that overexpression of mutant C-KRT74 likely enlarges Huxley cells by enhancing the KIF network, which in turn allows the hair shaft to be shaped more finely. This suggests that specific mutational events may interfere with normal levels of *KRT* and *KAP* gene expression and the corresponding regulatory mechanisms, resulting in wool fineness that differs from the usual situation. Thus, changes in the expression of specific genes (e.g., *KRT* and *KAP*) have different effects in different contexts, and further studies are needed to reveal their detailed molecular mechanisms and regulatory networks.

### 4.3. Analysis of Enrichment Results

Finally, enrichment of the candidate genes screened for the F vs. C group showed that among these 15 notable GO terms, the most relevant to hair follicle development were protein localization to plasma membrane and homophilic cell adhesion via plasma membrane adhesion molecules. These GO terms are involved in extracellular membrane-related processes, including cell adhesion and signal transduction, and play an essential role in hair follicle development and maintenance. The most relevant to wool traits is the positive regulation of G1/S transition of mitotic cell cycle, a GO term associated with cell division that can influence the rate and cycle of hair growth. The highest correlations with adipose metabolism are endoplasmic reticulum and ubiquitin-protein transferase activity. These GO terms are involved in cell metabolism and protein degradation that can regulate adipocyte differentiation and metabolic processes. The most relevant term for androgen metabolism is regulation of GTPase activity, which is involved in GTPase regulation and can regulate various signaling pathways, including the androgen receptor pathway. Of the 38 significant KEGG pathways, the most relevant pathways for hair follicle development and wool traits are the Regulation of actin cytoskeleton and the Phosphatidylinositol signaling system, both of which play a role in cell morphology and cytoskeletal organization and may be necessary for hair follicle formation/development and wool morphology. The pathways most closely linked to lipid metabolism are the Regulation of lipolysis in adipocytes, the Sphingolipid signaling pathway, and the Cortisol synthesis and secretion. These pathways are involved in the metabolic processes and energy balance of adipocytes, which may impact the accumulation and metabolism of subcutaneous adipose tissue. The pathway most closely associated with androgen metabolism is the PI3K-Akt signaling pathway, which is involved in androgen metabolism and cell signaling and may affect the formation of excessive hair and hair follicle development.

Enrichment of candidate genes screened for the C vs. F group showed that the 21 significant GO Terms with the highest relevance to hair follicle development were focal adhesion assembly and protein localization to plasma membrane. These GO Terms are involved in cell-membrane-related processes, such as cell adhesion and extracellular matrix–intracellular skeleton interactions, which are essential in hair follicle development. The most relevant to wool traits are protein localization to plasma membrane and regulation of platelet-derived growth factor receptor–alpha signaling pathway. These GO terms play a key role in cell signaling and differentiation and influencing hair morphology and growth rate. The most relevant to adipose metabolism are the CMP catabolic process and chloride transport, which are involved in cellular metabolism and transport and can regulate adipocyte differentiation and metabolic processes. The most relevant to androgen metabolism is GTPase activator activity. This GO term involves GTPase regulation and can regulate various signal transduction pathways, including the androgen receptor pathway. Among the 16 prominent KEGG pathways associated with hair follicle development are Axon guidance and Regulation of actin cytoskeleton and MAPK signaling pathway, which are involved in cell motility, the cytoskeleton, and signaling processes that play an important role in hair follicle differentiation and maintenance. Regulation of actin cytoskeleton, MAPK signaling pathway, and Arginine and proline metabolism are linked to wool traits. These pathways can affect hair growth and traits by influencing cell growth, the cytoskeleton, keratin structure, and other aspects. Related to fat metabolism are beta-Alanine metabolism, Pantothenate and CoA biosynthesis, and Phenylalanine metabolism, all of which affect lipid metabolism through different metabolic pathways and molecular mechanisms. Gap junction is associated with androgen metabolism, a pathway that involves intercellular signaling that regulates apoptosis and proliferation affecting the action of androgens. Combined with the studies of Wu et al. [57] and Ma et al. [94], this also confirms that the significantly enriched pathways PI3K-Akt signaling pathway and MAPK signaling pathway found in this study are associated with wool fineness.

### 4.4. Exploration of Factors Affecting Fineness

Genetic and physiological factors, including follicle morphology and condition, nutrient availability, hormones, and the developmental cycle of the wool, mainly influence the diameter of wool. The diameter of the wool determines how fine or coarse the wool is. The hair follicle’s morphology and growth state influence the wool’s diameter, shape, elasticity, and curl of the hair in relation to the underlying morphology and growth cycle of the hair. There is a correlation between the shape, elasticity, and curl of the wool and its diameter. In general, larger-diameter hairs are usually straighter; finer-diameter hairs usually have some curl; and smaller-diameter hairs that are flatter are usually more curved and curly. The shape usually refers to the underlying form of the hair, such as whether it is straight or curly. Elasticity refers to the ability of the hair to bounce back, while curl refers to how curly the hair is. Different factors may influence these traits, which in turn interact with and influence each other to shape the texture and characteristics of wool. Hair follicles and hairs are part of the skin and subcutaneous adipose tissue, which also plays an essential role in the growth and development of hair follicles. Adipocytes provide energy and nutrients, maintain the metabolic activity and proliferation of hair follicle cells, and release various fatty acids and signaling molecules to participate in the differentiation and repair process of the hair follicle. In addition, the signaling pathways within adipocytes can also influence hair health and growth by regulating the regulation of the nervous and endocrine systems. Thus, the metabolic processes and energy balance of adipocytes may affect the surrounding hair follicles and hairs, indirectly influencing wool development and traits. Studies have shown that the size and number of adipocytes are related to wool diameter [95], and they also play a non-negligible role in the development and traits of wool. Androgens play an essential regulatory role in the development of sheep hair follicles and have a significant impact on hair growth and development. Levels of androgens are related to the rate and stage of hair growth, and during development and adulthood, androgen levels can influence the size of wool diameter and wool traits. The pathway enrichment results associated with wool fineness and the Sankey diagram reveal that some genes are involved in multiple pathways (including hair follicle development, hair traits, fat metabolism, and androgen metabolism), which leads to speculation that they may be genes that affect hair fiber diameter. It is hypothesized that these genes (*PLCB2*, *PIK3CA*, *GNAI3*, *GRIA1*, *GRIA2*, *GRIA4*, *IKBKB*, *SKY*, and *PIP5K1B*) may be the ones that affect the formation of fine hairs. Moreover, these genes (*EFNA5*, *PAK3*, *PDGFD*, *PIP5K1B*, *FGF18*, *PIP4K2A*, *FGF2*, *PLCZ1*, *MAOA*, and *MAOB*) may be the ones that influence the formation of coarse hairs.

Therefore, we predict that these genes play an important role in the formation of wool fineness in sheep. Although some of these genes have been reported in sheep studies, the biological functions of these candidate genes in sheep wool fineness still need further validation.

## 5. Conclusions

In conclusion, this study explored the genetic variation, population structure, and selection characteristics of different woolly sheep populations using whole-genome resequencing. Population structure analysis showed that Hanzhong sheep were slightly segregated from other Asian sheep. We identified many genes, terms, and pathways related to hair follicle development, wool traits, fat metabolism, and androgen metabolism. We subsequently explored the relationship between fat metabolism, androgen metabolism, and hair fineness. This study confirmed previous studies, subdivided the genes related to wool fineness, and identified new genes that may affect wool fineness, such as *PLCB2*, *GNAI3*, *EFNA5*, and *PDGFD*. Our research results will help better promote wool improvement breeding, which is crucial for developing the fine wool sheep industry in China. It also supports the conservation and development of coarse wool resources.

## Figures and Tables

**Figure 1 animals-13-02944-f001:**
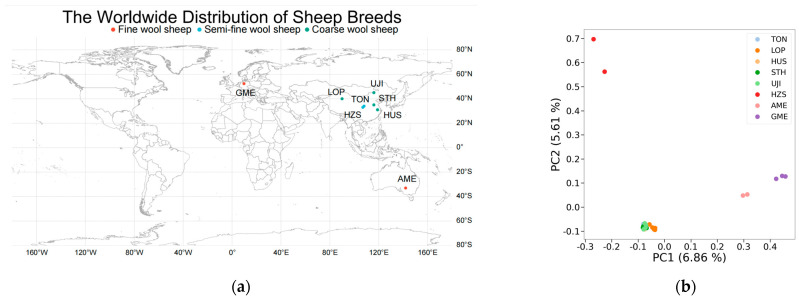
World distribution map of sheep breeds and population genetic structure analysis: (**a**) World distribution map of sheep breeds; (**b**) Principal component analysis (PCA); (**c**) Population structure analysis (different colors represent different components of ancestry); (**d**) Phylogenetic tree (N-J tree); (**e**) Cross-validation error.

**Figure 2 animals-13-02944-f002:**
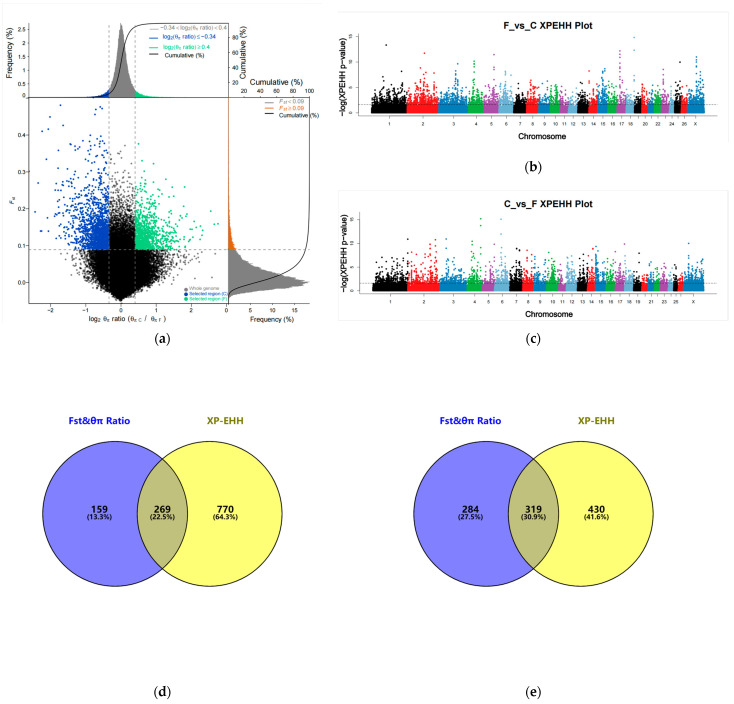
Selection signal analysis: (**a**) Fst and θπ Ratio selection elimination analysis plots; (**b**) Genome-wide distribution of XP-EHH (F vs. C); (**c**) Genome-wide distribution of XP-EHH (C vs. F); (**d**) Fst and θπ Ratio and XP-EHH screened for overlapping genes (F vs. C); (**e**) Fst and θπ Ratio and XP-EHH screened for overlapping genes (C vs. F).

**Figure 3 animals-13-02944-f003:**
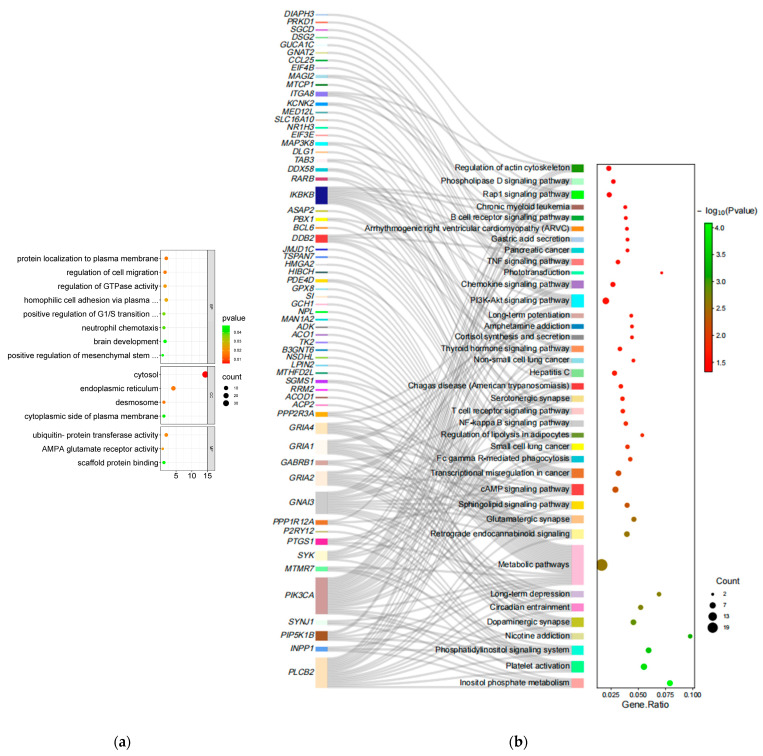
GO enrichment and KEGG enrichment results. (**a**) The most enriched GO Terms (F vs. C); (**b**) Sankey dot pathway enrichment (F vs. C); (**c**) The most enriched GO Terms (C vs. F); (**d**) Sankey dot pathway enrichment (C vs. F).

**Table 1 animals-13-02944-t001:** Information on the sheep populations in this study.

NO.	Breed	Abbreviation	Location	Photo	Size	Use	Wool Type
1	Australian Merino	AME	Europe	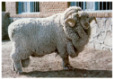	2	Wool sheep	Fine wool sheep
2							
3	German Mutton Merino	GME		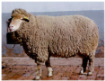	3	Wool sheep	Fine wool sheep
4							
5							
6	Tong Sheep	TON	Eastern Asia	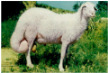	3	Wool and meat sheep	Semi-fine wool sheep
7							
8							
9	Hanzhong sheep	HZS		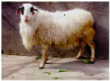	2	Wool and meat sheep	Semi-fine wool sheep
10							
11	Lop sheep	LOP		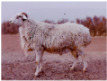	5	Wool and meat sheep	Coarse wool sheep
12							
13							
14							
15							
16	Hu Sheep	HUS		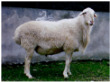	3	Wool and meat sheep	Coarse wool sheep
17							
18							
19	Small Tailed Han Sheep	STH		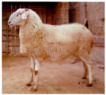	5	Wool and meat sheep	Coarse wool sheep
20							
21							
22							
23							
24	Ujimqin sheep	UJI		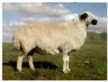	5	Wool and meat sheep	Coarse wool sheep
25							
26							
27							
28							

**Table 2 animals-13-02944-t002:** The distribution of SNP variants in the genome region.

Catalogue	SNP Numbers
Upstream	93,281
Stop gain (Exonic)	734
Stop loss (Exonic)	188
Synonymous (Exonic)	68,110
Non-synonymous (Exonic)	82,379
Intronic	8,123,336
Splicing	4227
Downstream	125,137
Upstream/downstream	2550
Intergenic	13,414,800
ts	14,501,815
tv	7,631,392
ts/tv	1.9
Total	22,133,207

## Data Availability

Data are available upon request due to privacy/ethical restrictions.

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
