# Peer review of "Whole-Genome Resequencing Reveals Selection Signal Related to Sheep Wool Fineness"

_animals, 2023, doi:10.3390/ani13182944_

Round 1

Reviewer 1 Report (New Reviewer)

Dear Authors, the topic of the paper is relevant because the

 However, the paper should be carefully revised before further evaluation.

1.     The major limitation is a study design. Sample sizes are too low (five animals in fine wool group, five sheep in semi-fine wool group, and 18 sheep in coarse wool group). The authors should provide justification and explanation why they used so small sample sizes in their research. The breeds, which were chosen for this study, are popular, well-known, and numerous and their representation in this work is seemed to be underestimated. In addition, I assume that the authors could find WGS data for these breeds in online databases and expand the sample sizes.

2.     I guess that Chinese Merino is a popular Merino derived breed created in People Republic of China. Could you please provide a comment why this breed was not included in the studied sample?

3.     No information of sampling strategy is provided. What was the age of studied sheep? Were the individuals within the same breed unrelated?

4.     L163-164 «Before conducting the analysis, AME, GME, TON, and HZS were clustered as F and LOP, HUS, STH, and UJI were clustered as C». However, PCA and ADMIXTURE analyses showed that Tong breed is clearly jointed the coarse wool breeds while Hanzhong breed was separated from other breeds. Besides two individuals of Hanzhong breed were quite differentiated from each other.

In this regard, the assignment of Tong and Hanzhong breeds to F cluster is not seemed to be methodologically correct and may affect the results of the study. I recommend removing these breeds from the analysis.

5.     How did the authors dissect the genes associated with fine wool formation (F vs. C) from those genes associated with coarse wool formation (C vs. F)? It is unclear and should be precisely described in Methods or Results sections.

6.     Table 1 should be revised. Column with Location should be corrected because it seems like Lop, Hanzhong and Tong breeds are of European origin, not Chinese.

7.     Phylogenetic tree (N-J tree) is missing in Figure 1.

8.     The Discussion may be revised by using shorter and more precise expressions.

9.     The photograph of studied sheep breeds would be attracting the attention of the potential readers.  

 Minor editing of English language required

Author Response

Thank you for your valuable comments. On the premise of ensuring that the content and structure of the paper remain unchanged, we have made every effort to improve the manuscript and have revised it accordingly. We sincerely appreciate your hard and patient work and hope that our revisions will be recognized by you. Thank you again for your insightful comments.

Reviewer 2 Report (New Reviewer)

This research provides interesting information. However, some very important changes need to be made in order to be considered for publication.

INTRODUCTION

Specific comments:

Line 87.- change "Eda" to "EDA".

Line 95-106.- restructure this sentence because it is only supported by one sentence [27].

MATERIAL AND METHODS

General comments: I recommend mentioning what were the criteria for selection and management of these animals. In addition, the statistical management performed for the variables evaluated, what significance value was considered (P<0.05), etc., should be specified.

RESULTS

General comments: I recommend increasing the size of the literals in the figures (Fig 1a; Fig 2a, b, c; Fig 3a, b, c, d).

DISCUSSION

In general, I recommend making an explanation of the findings observed in the research and try to explain what was the mechanism involved. In addition, it is advisable to write this section as the results were mentioned. For example in line 210, they mention "Genetic Variation and Population Genetic Analysis". However, they do not discuss or compare these results at the beginning of the section. Also, from lines 311-336 you do not compare or support the comments with any citations. Some sentences mentioned above would better belong in the "material and methods" section.

Much of this section compares gene association results with "Goats" are the same mechanisms even if they are different species?

I recommend further discussion of the methods "Fst and θπ Ratio" and "XP-EHH".

Line 347.- "hair follicle (HF)" is mentioned but in line 85 it was already defined.

Line 350.- "Edar" is mentioned but in line 67 it says "EDAR" Homogenize all the abbreviations in the document.

Line 405.- "androgen receptor (AR)" is mentioned in italics and on line 406 "AR", and "androgen receptor (AR)" is mentioned again on line 386. Adjust all repeated abbreviations throughout the document.

CONCLUSIÓN

I recommend restructuring this section and specifying the research findings.

Author Response

Thank you for your valuable comments. On the premise of ensuring that the content and structure of the paper remain unchanged, we have made every effort to improve the manuscript and have revised it accordingly. We sincerely appreciate your hard and patient work and hope that our revisions will be recognized by you. Thank you again for your insightful comments.

Reviewer 3 Report (New Reviewer)

The authors performed a whole-genome 110 resequencing to reveal the selection signatures of sheep wool fineness.

This an interesting study that provides useful information, but the manuscript is not yet to publication standards, hence revision is necessary before potential acceptance.

Major issues.

Simple summary. The simple summary is for reading by lay people, e.g., farmers, butchers, accountants, military personnel, etc., and certainly not scientists. So, the sentence ‘Fixation Index and Nucleotide Diversity Ratio (Fst & θπ Ratio), Cross Population Extended Haplotype Homozygosity (XP-EHH). Overlapping Single-Nucleotide Polymorphisms (SNPs) loci located in the top 5% of Fst & θπ Ratio and XP-EHH were screened for annotation’ is totally out of place. All the simple summary must be understood by the above lay people. So, the section must be rewritten from scratch.

Phenotypes. Who confirmed the phenotypes of the breeds assessed in the study?

Selection. The criteria for selecting the individuals that were samples must be clearly presented in detail.

Minor issues.

Subsection 3.3. Please present these results in a form of table.

Discussion. Please divide in smaller subsections to improve readability of the study.

Author Response

Thank you for your valuable comments. On the premise of ensuring that the content and structure of the paper remain unchanged, we have made every effort to improve the manuscript and have revised it accordingly. We sincerely appreciate your hard and patient work and hope that our revisions will be recognized by you. Thank you again for your insightful comments.

Round 2

Reviewer 1 Report (New Reviewer)

Dear Authors,

the paper was improved. 

Minor editing of English language required

Author Response

[Comment] The paper was improved. Minor editing of English language required.

Response: Thank you very much for your review and feedback on our manuscript. We are very pleased that you think our paper has improved. In response to your mention of the quality of the English language needing some minor editing, we carefully revised the paper to ensure accuracy and fluency of the language.

We started by reviewing the paper through Grammarly and Quillbot to correct any punctuation errors, grammatical mistakes, syntactic structures, and poor word choice. Then, we asked a teacher with excellent English skills to review the text for us again. We will make sure that the language is standardized, concise, clear and overall readable.

Thank you very much for your comments and suggestions, your professional opinion is very important to us. We have made every effort to rectify the situation and ensure that the article is even better in terms of language presentation (the changes have been corrected in the latest manuscript).

Thank you again for your review and valuable suggestions.

Reviewer 3 Report (New Reviewer)

The authors have improved the manuscript. However, I just noticed the recent publication of one or two relevant papers by other groups, publications which are topical to the subject of the article, so I ask to go through the literature and discuss their findings in relation to those of other authors.
Then, the manuscript can be accepted.

Author Response

[Comment] The authors have improved the manuscript. However, I just noticed the recent publication of one or two relevant papers by other groups, publications which are topical to the subject of the article, so I ask to go through the literature and discuss their findings in relation to those of other authors. Then, the manuscript can be accepted.

Response: Thank you very much for your feedback and suggestions. I understand that there are one or two recently published papers from other research groups related to the topic of our article following our submission. In our latest manuscript, we have refined our literature review and discussed the relationship between the findings of other authors and our findings. This will help to strengthen the relevance and accuracy of our article and also explore some new findings. We would very much welcome your guidance and suggestions to ensure that our article review is complete and persuasive.

We searched NCBI for literature related to wool fiber diameter, wool diameter, and wool fineness in the last three months and found three articles related to this study, which we compared and synthesized with our findings. We discussed these studies in depth and explored to get the support for some of the experimental protocols of this study (the theoretical basis for the selection of signals with a threshold of the first 5%), confirmed the function of some unknown genes in this study, and the role of some of the pathways. In addition to this, findings related to KRT and KAP genes were explored. This contrasts, complements or supports our study and enriches the discussion part of our literature.

Modify the following:

Wool fineness has a similar genetic basis in different breeds with low to medium heritability[57], thus genes associated with wool fineness may be found in regions of low genetic diversity. Therefore, in this study, the threshold of selection signal analysis was adjusted to the top 5% to avoid missing candidate genes related to wool fineness[58]. (Please see page 13, lines 383~387)

Through the studies of Wu et al.[58], Ma et al.[95], it was also confirmed that MYO3B and MACROD2 genes were associated with hair fineness and FST genes were associated with wool density in this study. Combined with the studies of Wu et al.[58], Ma et al.[95], we can learn that although KRT and KAP genes are the main structural proteins of animal hair, the expression of these genes is relatively low in the finer wool. This may be part of the developmental and structural regulatory mechanism of finer wool. However, the study by Liang et al.[9] reveals a special case that Overexpression of mutant C-KRT74 likely enlarges Huxley cells by enhancing the KIF network, which in turn allows the hair shaft to be shaped more finely. This suggests that specific mutational events may interfere with normal levels of KRT and KAP gene expression and the corresponding regulatory mechanisms, resulting in wool fineness that differs from the usual situation. Thus, changes in the expression of specific genes (e.g., KRT and KAP) have different effects in different contexts, and further studies are needed to reveal their detailed molecular mechanisms and regulatory networks. (Please see page 15, lines 491~504)

Combined with the studies of Wu et al.[58] and Ma et al.[95], it also confirms that the significantly enriched pathways PI3K-Akt signaling pathway and MAPK signaling pathway found in this study are associated with wool fineness. (Please see page 16, lines 555~558)

Thank you for your interest in our paper and the seriousness with which you reviewed it. We endeavor to revise our manuscript and synthesize the latest relevant studies to give more depth and breadth to our research. Please feel free to let us know if you have any other suggestions or requests. We look forward to working with you in order to bring our paper to its optimal contribution to the field. 

Thank you for your time and valuable feedback.

This manuscript is a resubmission of an earlier submission. The following is a list of the peer review reports and author responses from that submission.

Round 1

Reviewer 1 Report

The authors have divided eight sheep breeds into fine and coarse wool groups. Twenty-three genome, samples were sequenced and their SNPs were extracted to search for selection signals with different methods. The genes around the overlapping hits were reported and described by their function.

I suggest the manuscript for publication after modifications.

Notices:

Thirty-four times I felt I have to suggest partial deletions from the text, to explain general words in more detail.  I do not want to repeat them here, tey can be found in the attached pdf file.

The main things to be changed are:

Delete the 'analysis' word from the title.

H1 and H2 groups are not explained in the abstract. I  also suggest finding other names for better readability. Maybe it is better to call them F and C groups after the words 'fine' and 'coarse', respectively. That way, the readability improves all over the text. H1 and H2 are not intuitive names, hard to recode again and again. Find something better. It is not necessarily F and C.

Some resolutions of acronyms should be given earlier.

The authors complains about '...some studies have identified candidate genes that affect sheep wool fineness, but these genes often reveal only a certain proportion of the variation in wool thickness...'. I miss the number of genes identified by others, and the variation values they are referring to.

I also would like to read about that in the text, not just in the abstract.

The English language is fine, in spite of the fact, that several times I deleted words for better understanding. Nevertheless, I suggest a thorough re-read.

Reviewer 2 Report

Brief Summary

The present work deals with the search for selection signals related to wool fineness through the analysis of 8 breeds of sheep with fine wool, semi-fine wool and coarse wool (Australian Merino, German Mutton Merino sheep, Tong sheep, Hanzhong sheep, Lop sheep, Hu sheep, Small Tailed Han sheep and Ujimqin sheep). For this purpose, 28 animals were sequenced using illumina technology and only SNPs with a minimum depth of 2 were retained.

The authors found differences at the genomic level between the breeds used reflected in a principal component analysis and a population structure analysis.

General concept comments

The present work has serious shortcomings in the experimental development, which prevent reliable conclusions from being drawn from the results obtained. The authors have used 8 breeds, but each of them has a very low n-value. A total of 28 animals were studied. One of the breeds has only 2 animals. The total number of fine wool animals in the present work is 5. On the other hand, the animals from semi-fine wool breeds had to be added to these 5 animals in order to make a larger group on which to search for differences (H1, 10 animals). It is proposed to considerably increase the n for each of the breeds to a number that can be more representative of each group (n=25 as a minimum) and allow to determine that the differences found are a reflection of differential wool quality and not of differences due to the breed.

Specific comments:

Line 131: Authors should define better the libraries preparations (was used the Illumina kit? In any case must be referred and the company has to appear) and the sequenciation method. Was used NovaSeq 6000?

Line 151-152: Did you calculated the optimal K value? Should be included the optimal K and the method employed (x.e. In the Admixture software you can use the CV error to detect the optimal K)

Line 154-155: The cluster created incorporate breed of fine wool and semi-fine wool sheep in the same group so this H1 group cannot be defined as fine wool group.

Line 220: In the figure 1, the colors of the populations are different in the b) and d) sections. In order to facilitate the interpretation, the color of the populations has to be the same in this sections. In the section c) you should indicate the optimal K witch should be discussed after.

Reviewer 3 Report

Dear Authors, the idea to identify new markers associated with wool fineness is, of course, interesting, but using only 2, 3 or 5 animals per breed is unacceptable for study based on whole-genome resequencing, additionally in one group you have male and female and in other only female or male. In my opinion, this kind of research needs much more individuals, see the other papers, and based on only this number does not deliver reliable results, because observed results are probably associated with specific genetic features of these two sheep in one breed, and not specificity for all sheep population.  Moreover, we do not have any information if these individuals are related to themself, which is this kind of information would be vital.  Because obtained results based on this methodological approach seem to be unreliable so I won`t be commenting on the results and the discussion section. I`m very sorry but this kind of research needs much more individuals since wool fineness is quantity, not quality traits.